# Variation of Diatoms at Different Scales in the Brazilian Pantanal Basin

**Margaret S. Nardelli** [1,*], **André A. Padial** [2,3], **Denise C. Bicudo** [4], **Claudia M. d. S. Cordovil** [5] **and Silvio C. Sampaio** [1]

1. Programa de Pós-graduação de Engenharia Agrícola, Recursos Hídricos, Universidade Estadual do Oeste do Paraná, Cascavel CEP 85819-110, Brazil; silviocesarsampaio@gmail.com
2. Laboratório de Análise e Síntese em Biodiversidade, Departamento de Botânica, Programa de Pós-graduação em Ecologia e Conservação, Programa de Pós-graduação em Botânica, Universidade Federal do Paraná, Curitiba CEP 81531-990, Brazil; aapadial@gmail.com
3. Programa de Pós-graduação em Ecologia de Ambientes Aquáticos Continentais, Núcleo de Pesquisa em Limnologia, Ictiologia e Aquicultura (Nupelia), Universidade Estadual de Maringá, Maringá CEP 87020-900, Brazil
4. Núcleo de Ecologia, Instituto de Botânica, SIMA, São Paulo CEP 04301-902, Brazil; denisebicudo@gmail.com
5. Instituto Superior de Agronomia, Centro de Estudos Florestais CEF, Universidade de Lisboa, Tapada da Ajuda, 1349-017 Lisboa, Portugal; cms@isa.ulisboa.pt
* Correspondence: margaretseghetto@hotmail.com; Tel.: +55-045-99839-0077

**Abstract:** (1) Background: We analyzed the diatom community structure of the surface sediments, in three permanent ponds in the Pantanal of Mato Grosso, Brazil, to better understand how biota in these aquatic environments depend on structural connectivity and functional connectivity: (2) Methods: Ten samples sites were established in each pond, water and the sediment were taken during the flood period. Abiotic–biotic variables were determined and standardized; (3) Results: The three ponds presented acidic water and high concentration of nitrogen, with the highest acidity for Ferradura Pond (P1) and the highest trophic status index for Burro Pond (P2), but the greatest environmental variations occurred in Caracará Pond (P3). The variation in diversity between sites in the same pond is what contributes the most to gamma diversity. The most abundant species was *Aulacoseira italica* (Ehrenberg) Simonsen and the genus *Eunotia* Ehrenberg was the most representative in species. Ferradura Pond, there was a relationship between compositional and environmental dissimilarities with geographic distance, but there was no independent. Burro Pond, the relationship the compositional variation with environmental variables was not significant. Caracará Pond, there was a relationship of compositional dissimilarity both with geographical distance and with environmental; (4) Conclusions: The set of results suggests that the mechanisms that determine the metacommunity of each pond are different and that the environmental conditions and dispersion influenced the structure and composition. Since, diatom species were different between ponds, and ponds more eutrophic showed less diversity. The pH and oligotrophy were the main factors to maintain the greatest diversity of species of the genus *Eunotia* and the greatest abundance of *Aulacoseira italica*. Knowing the dynamics and structure of diatoms, which are at the beginning of the food chain, is essential for conserving, maintaining, or rehabilitating wetland ecosystems, such as the Pantanal, which is part La Plata river basin, which represents the second largest surface for water resources in South America and the Guarani Aquifer System, the biggest unified groundwater aquifer in the world.

**Keywords:** connectivity; diatom; environmental structure; metacommunity; surface sediments

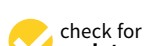

## 1. Introduction

Species vary over space and time according to a myriad of processes, describing variation in communities is now embedded in the metacommunity framework [1]. In aquatic ecosystems, local limnological variables, hydrological processes, and habitat connectivity

are the major determinants of species variation [2–4]. On the other hand, species variation can also be generated through stochasticity, for instance, in probabilistic and limited dispersal, which is well described in Hubbell's Neutral Theory [5].

It is now a consensus that the relative importance of mechanisms above mentioned depends on the spatial scale in which communities are studied [6–8]. For instance, at large spatial scales, even microorganisms with large dispersal ability can present some dispersal limitation [9–13]. At local scales, priority effects, micro-habitat differences, and biological interactions take place as relatively more important mechanisms to explain spatial and temporal differences in communities [11,14–21].

Diatoms are microalgae commonly used as bioindicators of water quality and environmental wealth of aquatic bodies, given their quick response to environmental variation [22]. Diatoms, reproduce primarily via asexual division with rare instances of sexual reproduction. Some species may divide once per day in optimal environments. Importantly, diatom valves are typically well preserved in the ponds' sediments [23]. It is also reported that dispersal related mechanisms have a role in determining differences of local diatoms communities [24–27]. In this sense, describing scales of the spatial variation and main correlations of diatoms community variation is the core to reveal the most important ecosystem processes determining aquatic metacommunities.

In addition to the known temporal variation, there is the spatial variation, floodplains like the Pantanal, have a high environmental heterogeneity in aquatic ecosystems, ranging from microhabitats to large scales [28–30], and it include the habitats colonized by species, such as diatoms. Following this line, species can vary within and between such habitats, identifying how variation occurs may reveal the main ecological processes determining diatoms.

Historically, ecologists debate whether it is true that everything is everywhere, but the environment selects [31–33] and there is dispersion to isolated spaces (e.g., ponds), with the success of immigration and colonization, with different biogeographic and biodiversity levels, depending primarily on the organism's ability to disperse [34]. Immigration of new species varies between scales, ecosystems, and organisms; however, what types of place are occupied by which species? Is very likely that due to the richness and structure of the species both by biogeographical factors as well as by local characteristics and connectivity forming a conundrum of likely underlying causes [35]. For instance, a watershed is a landscape unit connected through the flow of water among its sources all along its mouth, and can be assessed through their mosaic composition, whether composed by corridors, patches, matrices, or subunits within in watershed for planning, management, and analytical purposes [36].

Many are the responses projected with focus on wetlands around the world. However, water resources will not be sustainable or sufficient unless other indirect and direct drivers of change are addressed and do not lead to a reduction in the services provided by wetlands. Wetland studies are extremely important, because wetland do more than act as water filters, providing flood and erosion control, they also sustain plants and animals in the watershed. Studies of the diatom community at local scales add to this importance in wetlands, since first we have to understand how the dynamics of this community is structured in a micro-habitat, and then it can be applied in larger spatial scales. Knowing the dynamics and structure of diatoms, is essential for conserving, maintaining, or rehabilitating wetland ecosystems, and the Pantanal is part La Plata river basin, which represents the second largest surface for water resources in South America and the Guarani Aquifer System, the biggest unified groundwater aquifer in the world [36,37].

In this study we analyzed the structure of the diatom metacommunity in three permanent ponds of the Pantanal of Mato Grosso State in Brazil, to understand how species variation occurs at different scales. We aimed to better understand how biota in these aquatic environments depends on the geographic distribution of water bodies (structural connectivity) and effective movements between species and bodies of water (functional connectivity). Therefore, this study will contribute to answer to the following questions

and the concepts in metacommunities that will guide the hypotheses if the premises are true:

(1) Do communities differ more than expected at random levels between local Pantanal habitats (i.e., ponds), or the difference is just a mathematical artifact?

(2) If differences exist, which species characterize these differences? Does the different species composition a consequence of the environmental characteristics of the ponds?

(3) Is there a homogeneous or heterogeneous spatial composition in the ponds?

(4) Which is the scale that most contributes to the total diversity of diatoms: the samples made at each site, the compositional variation within the pond, or the compositional variation between different ponds? Which of these contributions are the greater, and which are less than expected by a null model?

## 2. Materials and Methods

### 2.1. Study Area

The area of the Pantanal complex, with 250 km$^2$, is located between parallels 15° to 22° south latitude and 55° to 58° west longitude, with double climatic seasonality, a predominant one with intense rainy season (October–April), with about from 70 to 80% of average annual precipitation and another (May–September) with physiological drought [38]. The annual average temperature oscillates around 25 °C (34 °C and 15 °C), and the precipitation 1400 mm, with variation between 800 and 1600 mm [38,39]. For data collection, the choice of the ponds was based on two principles: be a permanent pond and be connected to one of three different rivers in the Pantanal Basin.

- Ferradura Pond (P1) connects with the Cuiabá River; has an average width of 300 m, with an approximate length of 1200 m; the pond depth is of 270–650 cm; is located 107 km from the city of Cuiabá. The Cuiabá River is impacted by the discharge of sewage, as well as by the fish farming tanks that discharge their effluents with high levels of nitrogen [29].

- Burro Pond (P2) connects with the São Lourenço river; has an average width of 1000 m and 5000 m in length and a depth of 140–280 cm; is located 173 km from P1. The São Lourenço River is influenced by agricultural activities and also by heavy metals—mainly mercury due to gold mining activities [29].

- Caracará Pond (P3) connects with the Paraguay River, has 3000 m width and 3600 m length, has a depth of 120–290 cm, is located 7 km from P2. The Paraguay River is influenced by agricultural, livestock and heavy metals activities—mainly mercury due to gold mining activities [29]. Sampling locations, geographical coordinates of the three ponds listed in Table 1.

### 2.2. Experimental Design

Sample collections were performed in February 2015, in a flow line from the river to the pond. Ten sites of each pond (30 total samples) were designated, with superficial sediment samples (SSF), performed with an Ekman collector, in the first 2.0 cm of bottom of each pond. Concomitantly, water was collected from the sub-surface of the ponds, and both samples collections were kept under refrigeration.

### 2.3. Diatoms

Sub-samples of the SSF samples (0.5 g) were oxidized according to the adapted method [40] and permanent slides were mounted (1 mL of oxidized), fixed with Naphrax for quali-quantitative analysis. For the qualitative analysis, an image capture microscope was used, and for the identification of diatoms, the resources of classic works, ecological floras and specific articles in the area were used. The classification system of Round [41]. The codes of the diatom species were assigned according to the OMNIDIA software [42]. For quantitative analysis, slide counts were performed, increasing by 100X until reaching 400 diatom valves to verify the percentage of the relative density of each sample [40]. Tables were generated with the species by locations and relative abundance values.

**Table 1.** Geographic coordinates of the sampling sites corresponding to the three ponds.

| * Ferradura Pond (P1) | | | * Burro Pond (P2) | | | * Caracará Pond (P3) | | |
|---|---|---|---|---|---|---|---|---|
| Site | Latitude | Longitude | Site | Latitude | Longitude | Site | Latitude | Longitude |
| 1 | 16°31′35″ S | 56°23′26″ O | 11 | 17°50′24″ S | 57°23′53″ O | 21 | 17°53′32″ S | 57°27′55″ O |
| 2 | 16°31′34″ S | 56°23′25″ O | 12 | 17°50′22″ S | 57°23′44″ O | 22 | 17°53′42″ S | 57°27′19″ O |
| 3 | 16°31′32″ S | 56°23′26″ O | 13 | 17°49′18″ S | 57°24′05″ O | 23 | 17°53′11″ S | 57°27′25″ O |
| 4 | 16°31′29″ S | 56°23′26″ O | 14 | 17°49′00″ S | 57°23′49″ O | 24 | 17°52′32″ S | 57°27′29″ O |
| 5 | 16°31′23″ S | 56°23′32″ O | 15 | 17°48′46″ S | 57°23′18″ O | 25 | 17°52′11″ S | 57°27′45″ O |
| 6 | 16°31′22″ S | 56°23′35″ O | 16 | 17°47′30″ S | 57°23′28″ O | 26 | 17°51′32″ S | 57°27′17″ O |
| 7 | 16°31′21″ S | 56°23′41″ O | 17 | 17°46′40″ S | 57°22′55″ O | 27 | 17°51′07″ S | 57°27′40″ O |
| 8 | 16°31′19″ S | 56°23′47″ O | 18 | 17°46′16″ S | 57°22′39″ O | 28 | 17°50′51″ S | 57°27′46″ O |
| 9 | 16°31′24″ S | 56°23′57″ O | 19 | 17°46′09″ S | 57°22′37″ O | 29 | 17°50′34″ S | 57°27′44″ O |
| 10 | 16°31′25″ S | 56°23′55″ O | 20 | 17°45′47″ S | 57°22′28″ O | 30 | 17°50′29″ S | 57°27′54″ O |

* Distances between the largest city (Cuiabá) and the first pond and between the ponds: Cuiabá city and P1 = 107 km; P1 and P2 = 173 km; and P2 and P3 = 7 km).

### 2.4. Physical and Chemical Variables

The physical and chemical parameters of the water (temperature, pH, dissolved oxygen and turbidity and conductivity, total dissolved solids, and depth) were obtained with the multiparameter sounder (Horiba U50). The levels of total phosphorus (TP) and total nitrogen (TN) in the water were analyzed by method of Valderrama [43] and the analysis of chlorophyll a by method of Marker [44]. The trophic state of the water index was established according to Lamparelli [45], adopting values of classification of eutrophication for lentic environment, for chlorophyll a and TP.

### 2.5. Data Analysis

For the abiotic data (physical and chemical), as well as the indicator species (relative species abundance), a comparison was made of the data obtained by recording the results of the values, maximum (Max), minimum (Min) values, coefficient of variation (CV%), and standard deviation (SD). For each pond, mean values (± standard deviation) the alpha diversity, abundance, dominance, and equitability of species were calculated based on the indices: Shannon H, Simpson 1-D, Fisher-alpha, Menhinick, Berger–Parker, and ANOVA tests show differences among ponds.

To obtain the answers to the objectives, a PERMANOVA [46] was first applied using the Bray–Curtis dissimilarities in data values of the sites collected inside the ponds as units to represent the local composition of the pond, in relation to the community structure (Goal 1).

To describe a possible environmental variation between the ponds, the environmental variables were previously standardized, applying a PERMANOVA. Additionally performing main coordinate ordering analysis PCoA [47], to see which pond differs most among the three ponds (significant differences), both in composition and in environmental variables. In sequence were applied, Bray–Curtis and Euclidean distances from the matrix of abundance composition of the communities and previously standardized environmental variables. In case of differences, a test of the species indicator value was applied IndVal [48] In case of differences, a test of the species indicator value was applied to characterize the compositional differences (Goal 2).

To assess if communities differ also in compositional variation, a permutation test was applied after a betadisper approach using Bray–Curtis dissimilarities [49]. The result of this permutation indicates if the metacommunity of one pond is more variable than the community of another pond. In case of differences, the most variable pond will be identified with the same PCoA ordering described above. The variation is estimated as the mean distance from the sites to the centroid of the composition distribution in the PCoA multivariate space. This analysis was also performed with Euclidian distances based on standardized environmental variables to describe if ponds change in environmental heterogeneity (Goal 3).

In addition, a Mantel test [50] was applied to data of each pond separately, to evaluate if inside each pond there is a continuous spatial structure in the community, correlating the Bray–Curtis dissimilarity matrices between sites in each pond, with the geographical distances between sites in each pond. To calculate the geographic distance matrix, Euclidean distance was applied to the geographical coordinates of the sites. Although the ponds are not perfectly linear, these distances still represent sites closer to each other, and the most likely relationship amongst sites (Figure 1 and Table 1). In this case, if there is spatial structuring in the composition, it will also be evaluated if there is a spatial structure in the environmental variables, and also if there is a correlation between compositional dissimilarity and environmental dissimilarity, regardless of the distance between sites. If this is the case, it indicates that the community varies locally depending on environmental conditions. For this, the partial Mantel test will be used (Goal 4). Finally, it was evaluated which scale contributes the most to the total diversity of this diatom metacommunity: if it is the local diversity of each site (alpha), if it is the variation in the diversity between sites of the same pond (beta1), or if it is the variation in the diversity between ponds (beta 2). These quantities were estimated by the additive metacommunity partition [51] and their values compared with a null model, to assess which quantities are greater or less than would be expected by a permutation test that assumes that all species could be recorded in all sampling units. For this analysis, only the presence and absence data were used to partition the species richness (Goal 5).

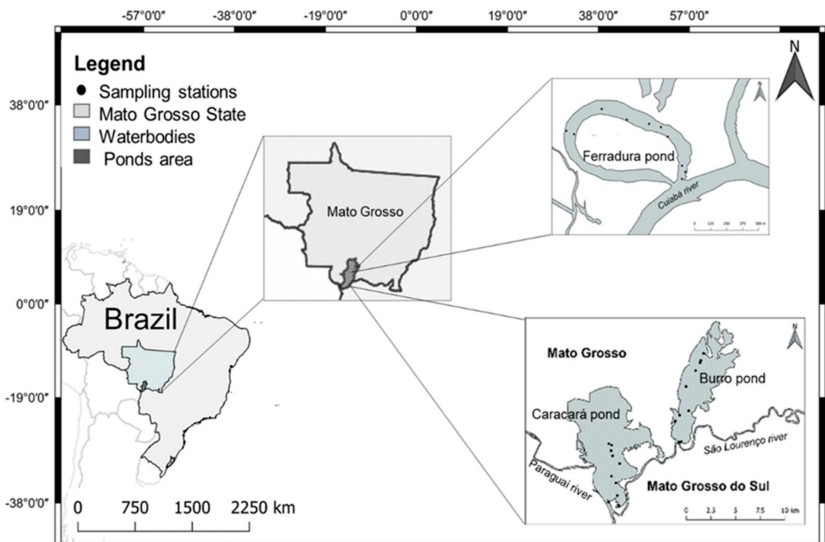

**Figure 1.** Map of Brazil with the location of the three ponds, marking the 10 sampling sites of each pond: Ferradura Pond (P1) located between the coordinates 16°31′24″ S and 56°23′40″ W (River Cuiabá), Burro Pond (P2) located between the coordinates 17°45′46″ S and 57°23′44″ W (River São Lourenço) and Caracará Pond (P3) located between the coordinates 17°50′33″ S and 57°27′52″ W (River Paraguay).

### 3. Results

A total of 119 species of diatoms distributed in 31 genera were reported from the qualitative analysis of the sampling of three permanent ponds located in the floodplain of the "Pantanal Mato-Grosso". In general, the most representative genus in abundance was *Aulacoseira* Thwaites. The most abundant species was *Aulacoseira italica* (Ehrenberg) Simonsen. Considering species richness, the genus *Eunotia* Ehrenberg was the most representative in species, with 39 taxa recorded.

According to the Mennhinick Index, which considers diversity according to the registered numbers, a higher index (2.626) was observed for P1, due to the fact that this pond has the highest number of taxa (83) in relation to P2 (72 taxa) and P3 (78 taxa). Along with Fisher's index, which assumes that abundance follows the distribution of the log series, it also resulted in a higher index (21,500) for P1 with the highest number of valves per ml (763,060) and for P3 with the lowest number of valves per ml (706,322). The Berger–Parker dominance index, which uses the measure of the numerical importance of the most abundant as a parameter, also generated greater value for P1. However, according to the Simpson and Shannon index, which consider diversity in relation to uniformity, it revealed the greatest diversity for P2, even with the smallest number of species, but with the best distribution of individuals (Table 2).

**Table 2.** Mean values (±standard deviation) of diversity indexes (see methods) for the sampling sites in each of the three ponds studied. ANOVA tests show differences among ponds. The indexes for each pond are also shown considering all sampling sites in each pond pooled.

| Index | Ferradura Pond | Burro Pond | Caracará Pond |
|---|---|---|---|
| Species richness | **83** | 72 | 78 |
| Simpson 1-D | 0.810 | **0.923** | 0.902 |
| Shannon H | 2.580 | **3.126** | 3.099 |
| Menhinick | **2.626** | 2.276 | 2.466 |
| Fisher-alpha | **21.500** | 17.790 | 19.780 |
| Berger–Parker | **0.397** | 0.189 | 0.205 |

Note: in bold the highest values the of indexes.

Goal 1: the analyzed metacommunity has coherence and a non-random structure [52]. Accordingly, ponds differ from each other more than would be expected by a null model, according to PERMANOVA (Figure 2). Only the classification of ponds explains 24% of the spatial changes in community composition (F = 8.94; R2 = 0.24; $p < 0.001$). It is visible in a PCoA diagram that there is little compositional superposition, however, the pond directly associated with the Cuiabá River (P1) is the one that most differs in species composition. It is also the pond that has the greatest numerical richness (83) in relation to the other two ponds.

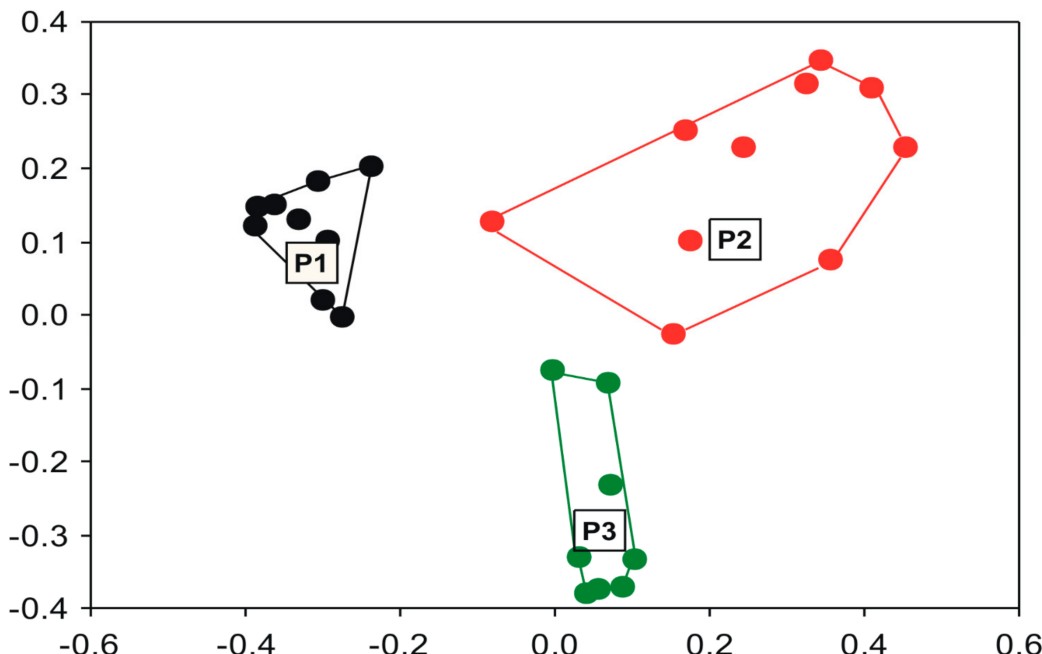

**Figure 2.** Variation in community structure in response to the spatial variables of Ponds 1, 2, and 3 in the Pantanal Basin.

Goal 2: Ponds also differ among each other for the set of environmental variables, with an explanation power of 27% (PERMANOVA F = 10.51; R2 = 0.27; $p < 0.001$). The difference between all ponds is also visible in a PCoA scatterplot, but with a continuous change between ponds 1, 2 and 3 compared to the differences in species composition (Figure 3, see also Figure 2).

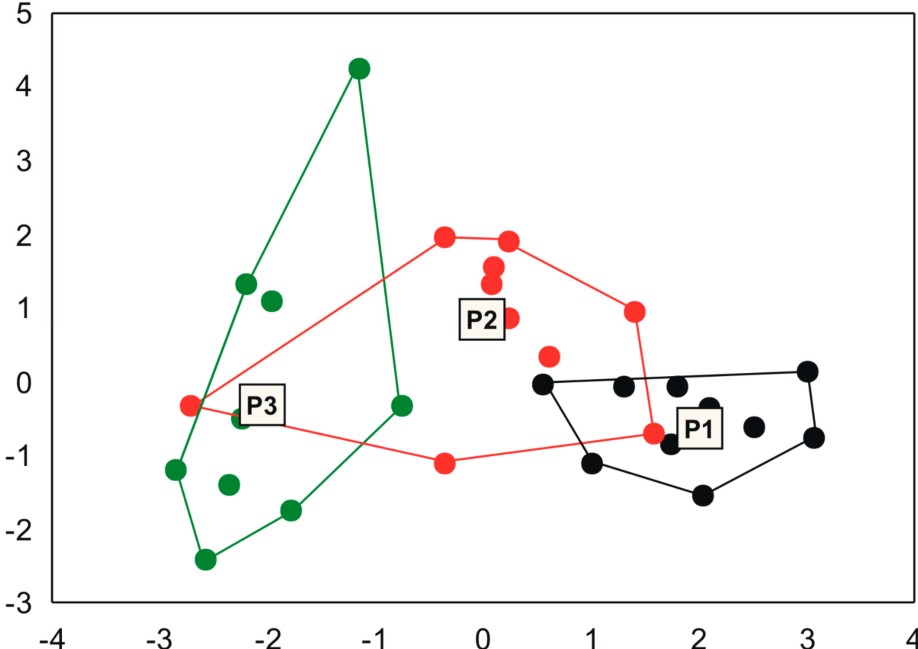

**Figure 3.** Variation in environmental conditions between Ponds 1, 2, and 3 in the Pantanal Basin.

The highest concentrations of total nitrogen were observed at P1. The most conspicuous difference in environmental variables occurred between the sampling sites of P3; however, P2 had a high variation in turbidity between the 10 sampling sites (Table 3).

**Table 3.** Values minimum (Min), maximum (Max), standard deviation (SD) and coefficient of variation (CV%) for the three ponds under study.

| Site | | Depth | T °C | pH | TDS ug·L$^{-1}$ | Cond µS·cm$^{-1}$ | TU | N ug·L$^{-1}$ | P ug·L$^{-1}$ | DO | TSI |
|---|---|---|---|---|---|---|---|---|---|---|---|
| P1 | Min | 320 | 28.2 | 5.8 | 29 | 0.04 | 12.1 | 1612.8 | 20.3 | 27.7 | 49.8 |
| | Max | 650 | 29.1 | 6.4 | 31 | 0.05 | 27.1 | 2822.4 | 36.8 | 41.4 | 51.4 |
| | SD | 111 | 0.3 | 0.2 | 0.7 | 0 | 4.2 | 395.1 | 5.4 | 5.3 | 0.5 |
| | CV (%) | 25 | 0.9 | 3.4 | 2.3 | 2.17 | 25.3 | 18.8 | 20 | 15.4 | 1 |
| P2 | Min | 160 | 28.1 | 5.8 | 24 | 0.04 | 0 | 1478.4 | 16.6 | 29.7 | 56.3 |
| | Max | 380 | 29.5 | 6.7 | 32 | 0.05 | 47.2 | 2419.2 | 49.3 | 55.4 | 59.1 |
| | SD | 59 | 0.4 | 0.3 | 2.9 | 0 | 15.3 | 276.7 | 8.3 | 7.5 | 0.7 |
| | CV (%) | 27 | 1.4 | 4.4 | 10.2 | 9.86 | 70.3 | 15.1 | 26.4 | 18.2 | 1.2 |
| P3 | Min | 120 | 29.2 | 5.9 | 23 | 0.04 | 5.5 | 1478.4 | 13 | 48.6 | 53.7 |
| | Max | 270 | 30.2 | 6.8 | 38 | 0.06 | 61.6 | 2688 | 41.9 | 93.3 | 56.7 |
| | SD | 48 | 0.3 | 0.3 | 4.8 | 0.01 | 13.4 | 442.7 | 10.2 | 13 | 1.1 |
| | CV (%) | 25 | 0.9 | 4.7 | 17 | 17.14 | 34.7 | 22.7 | 42.7 | 20.7 | 1.9 |

Note: Depth collect (cm); Water temperature (T °C); Total dissolves solids (TDS ug·L$^{-1}$); Electrical Conductivity (Cond µS·cm$^{-1}$); Turbidity (TU); Dissolved oxygen (DO%); Total Nitrogen for water (N µg·L$^{-1}$); Total phosphorus for water (P µg·L$^{-1}$); Trophic status index (TSI), Ferradura Pond (P1), Burro Pond (P2), Caracará Pond (P3).

In a coherent way, there were several indicator species for each pond. The list of indicator species with a significant indicator value considering a permutation test for the three ponds was longer for P3 with 14 indicator species (Table 4).

**Table 4.** Species with significant indicator value for all Ponds. The codes of the diatom species were assigned according to the OMNIDIA software.

| P1 | | | P2 | | | P3 | | |
|---|---|---|---|---|---|---|---|---|
| Species Code | Indictor Value | *p* | Species Code | Indictor Value | *p* | Species Code | Indictor Value | *p* |
| AUIT | 0.81 | 0.001 | AAMB | 0.90 | 0.001 | SGOU | 0.99 | 0.001 |
| EMET | 0.67 | 0.001 | AUSI | 0.90 | 0.001 | EDMG | 0.98 | 0.001 |
| EDNT | 0.60 | 0.002 | ETFG | 0.85 | 0.001 | ECUT | 0.66 | 0.008 |
| DCOF | 0.45 | 0.016 | AUVE | 0.81 | 0.002 | PSP2 | 0.60 | 0.002 |
| PACR | 0.45 | 0.048 | ASP2 | 0.80 | 0.001 | PHUC | 0.60 | 0.001 |
| EFLX | 0.40 | 0.030 | AUMN | 0.70 | 0.001 | EREL | 0.59 | 0.001 |
| EYBE | 0.40 | 0.026 | AUPU | 0.70 | 0.001 | GNAV | 0.59 | 0.003 |
| NAIK | 0.40 | 0.026 | AUGR | 0.60 | 0.008 | EMAI | 0.53 | 0.003 |
| ACOP | 0.36 | 0.031 | DSTE | 0.58 | 0.003 | ISPL | 0.50 | 0.005 |
| SRUD | 0.34 | 0.038 | AUHE | 0.55 | 0.006 | UULN | 0.42 | 0.014 |
| - | - | - | EPAP | 0.40 | 0.031 | GYAC | 0.40 | 0.033 |
| - | - | - | SLDB | 0.40 | 0.025 | PSP1 | 0.40 | 0.027 |
| - | - | - | EFRM | 0.36 | 0.025 | EURS | 0.35 | 0.036 |
| - | - | - | - | - | - | FFRA | 0.35 | 0.049 |

Goal 3: Ponds also differed in compositional variation (F = 8.78; *p* = 0.002), with P2 being the most variable considering the average distance to the centroid of distribution in a PCoA scatterplot, followed by P3 and P1 (see the size of species distribution clouds in Figure 2; Average distance to the centroid: P1 = 0.252; P2 = 0.430; P3 = 0.292). Environmental heterogeneity was also different among ponds (F = 3.16; *p* = 0.042). However, the compositional variation was not completely coherent the environmental heterogeneity of ponds. Although P1 was also the pond with the lowest environmental heterogeneity (average distance to the centroid: 1.831) and compositional variation (see above), P3 had the highest environmental heterogeneity value (2.793) but not compositional variation (results above), followed by P2 (2.202), which had intermediate environmental heterogeneity (Figure 3) and also had intermediate absolute values of environmental variables.

Goal 4: There were evidences for spatial structuring in the composition of the three ponds (Mantel's r and *p*-values for Pond 1, 2 and 3, respectively: r = 0.366; *p* = 0.021; r = 0.364; *p* = 0.004; r = 0.475; *p* = 0.007). Sites within ponds also shown spatial autocorrelation in environmental variables in P1 (r = 0.505; *p* = 0.002); but not in P2 (r = −0.083; *p* = 0.725) and Pond 3 (r = 0.137; *p* = 0.180). The relationship of compositional dissimilarity with environmental dissimilarity, controlling for geographical distance, was not significant for P1 (r = 0.139; *p* = 0.253) and P2 (r = −0.291; *p* = 0.890), but was significant for P3 (r = 0.5603; *p* = 0.008).

Goal 5: The additive diversity partitioning indicated that, in absolute terms, the variation in diversity between sites within the same pond (beta 1) mostly contributed the gamma diversity. However, this portion was lower than expected by the null model, indicating that the variation within the pond is relatively low when compared to the coexistence of species at each site (alpha) and to the compositional change between ponds (beta 2). In fact, both alpha and beta 2 diversity portions were significantly higher than the expected by the null model (Table 5).

**Table 5.** Additive diversity partitioning showing the mean contribution of species in each site withing each pond (alpha), variation among sites within each pond (beta 1) and variation among ponds (beta 2). Values for observed and expected by null model are shown, as well as a standardized effect size and *p* value.

| | Observed Contribution | Standardized Effect Size | Expected Contribution | *p* |
|---|---|---|---|---|
| Alpha | 27.5 | 18.195 | 22.666 | 0.001 |
| Beta 1 | 50.167 | −14.182 | 70.378 | 0.001 |
| Beta 2 | 41.333 | 10.778 | 25.956 | 0.001 |

## 4. Discussion

The Pantanal lives in constant transformation, contemplates periodic floods, and supplies countless permanent and non-permanent ponds, as well as in physical scales, storage of water on, and below ground, because they are not physically independent, since groundwater interacts with surface water; for example, the Guarani Aquifer System, which is under increasing stress, and is being threatened with pollution and contamination [36,37]. Therefore, in period of flood, ponds may have a greater connection than in dry periods (not analyzed). Whereas, spatial variations in the structure and composition of communities are largely influenced by the seasonal flood pulse [53], as well as by other predictors who have significant relationships with diatoms, such as pH, longitude, annual temperature, and precipitation [25]. However, they can vary between spatial systems and scales, when the degree to which the dispersion limitation overlaps with environmental filtering process [18] and comes to depend on specific ecological characteristics of each species [54].

For Ferradura Pond (P1), the continuous influence on alteration of the communities may be due to the mass effects of the dispersion, or to the effects of the environmental structure. Paes and Blinder [55] report that places with a greater number of species may have characteristic environmental properties, such as greater diversity of habitats, bigger area, species with greater range of distribution, among others. Otherwise, Remmer et al. [26] report that beta diversity is greater in small lakes and decreases with the raise of the lake area, and that deeper ponds are less influenced than shallower ponds and the dispersion limitation generally increases with the raise of spatial distance between locations [18]. In addition, diatoms preferentially respond to trophic gradients [4], and metacommunity, made up of ponds with biogeographic restriction, it is reported that its local communities are highly resistant to invasion, unless there are significant disturbances [56]. *Eunotia* group and also *Aulacoseira italica* were prominence species, in P1. The *Eunotia* group is cited as the most diversified genus in acid and wetland environments [57–60], *Aulacoseira italica* is not a common species [61–63] and when found in greater abundance is in acid and oligotrophic environments [64–66].

At Burro Pond (P2), it is likely that only the effects of mass dispersion explain the compositional variation. The pond is considered the meeting site of the other rivers, facilitating the sustained dispersion in the pond. Even with the heterogeneous environment, the population's existence is supported by immigration, assuming that the dispersion is modest due to the changes resulting from the floods. Moreover, local species differ from other ponds, thus supporting a metacommunity [1]. As noted, species present different competitive skills, but remain in sub-optimal or even unfavorable habitats [67] rescued by recurrent immigration [68]. These source-sink dynamic mass effects modify species diversity [67] and consequently affect the structure and dynamics of the community [68]. The highlight for this pond (P2) was the largest number of species of the genus *Aulacoseira*. The ecological preference of the *Aulacoseira* show that the trophic gradient is the main driver of species distribution [66,69], and many species of the genus are typical of the mesotrophic for eutrophic environments [63,66,70–73].

For the Caracará Pond (P3), it is likely that both the effects of dispersion and environmental selection explain in a complementary way the compositional variation. It is the most environmentally heterogeneous pond, with a large variation among its 10 sampling sites. Studies suggest that the environmental filtering process is the main structuring factor for diatom communities [11,74,75], modeled particularly by the availability of nutrients [21] and widely influenced by the seasonal flood pulse [53], with dispersion being second structuring factor, highlighting the importance of also considering processes related to dispersion in the interpretation of diversity patterns. However, studies show that both processes, dispersion and filtering, can be of equal equivalence [76]. Pond 3 was the one presenting the most indicator species. Local communities within a metacommunity may result, according to Leibold et al. [77], in several spatial dynamics, altering the diversity of local species, both directly and indirectly, and can go back and change characteristics of the regional biota. Since that diatom community structures are also the result of filtering and spatial processes [54].

The diversity of each site in a pond, as well as the variation between ponds, was significant. Assuming that the dispersion is modest, and ponds maintain a metacommunity and not a homogenized local community, and if the dispersion were extremely low, it would depend fundamentally on stochastic processes and local interactions (species/environment). Studies report that the increase in alpha diversity occurs with the increase in dispersion rates [1,78], but if it is maintained eventually through the immigration of source populations, in this case there is an increase in alpha diversity and consequently there is a reduction in beta diversity [79]. However, the variation in diversity between sites in the same pond (beta 1) is what contributes the most to gamma diversity. Dispersal rates can have significant effect to species diversity and ecosystem stability [79], in which richness and stability are maximized at intermediate dispersal [80].

On the other hand, if even higher dispersion rates can make a difference if this effect occurs simultaneously for many species of the metacommunity, then the composition of the local communities will be homogenized between the spots. In this case, the beta diversity decreases, and the alpha of the local communities will approach the gamma diversity of the metacommunity. Moreover, if the dispersion rates are extremely high, it can be considered that the spatial fragmentation of the habitat is irrelevant for the species, and in this case, what we call a metacommunity is effectively just a local community [79].

As it becomes increasingly evident that human actions are exercising ever greater control over the conditions and processes that allow our existence, diatoms have proven to be extremely powerful indicators to explore and interpret many ecological problems. Thus, applied studies based on diatoms are tools that closely meet the general expectations of the environmental managers [81], as well as seeking to introduce environment related topics into projects and joint actions along transboundary water systems [36]. Studies of the diatom community at local scales are extremely important, because first we have to understand how the dynamics of this community is structured in a micro-habitat, and then it can be applied in larger spatial scales. Large wetlands may also be a union of several

smaller wetland types and are found around world. In this case, we need to start to study the basics to understand the functioning of any wetland.

## 5. Conclusions

Our results reveal that the processes related to the environment and dispersion are in control of the structure and composition of the diatom community in wetland. According to the data analyzed, it was particularly evident that different species responded to different limitations or restrictive environmental factors, with abundance of species that are more adapted to the environment.

Our studies reveal that environments that are more degraded have less local diversity and, because the areas maintained by the rivers receive an eutrophication load, there may be a change in the structure of the diatoms. Ponds that are more eutrophic showed less diversity and the pH and oligotrophy were the main factors to maintain the greatest diversity of species of the genus *Eunotia* and the greatest abundance of *Aulacoseira italica*. However, moderate dispersion also maintains species diversification, due to the fact that these ponds have a connection to a different river that supplies it.

Summing up, the ponds have unique dynamics, there are differences in the compositional variation, in the mass effect processes, and in the different indicator species for each pond. Knowing the dynamics and structure of diatoms, which are at the beginning of the food chain, is essential for conserving, maintaining, or rehabilitating wetland ecosystems, such as the area of our study. The Pantanal is part of La Plata river basin, which represents the second largest surface for water resources in South America and the Guarani Aquifer System, the biggest unified groundwater aquifer in the world. These results show that even environments that are of the same type (wetland) are unique and have differences from each other, so these areas cannot be generalized. Considering the context of the environmental fire destruction that the Pantanal suffered (2020), the results obtained in this study are of utmost relevance to raise new research for the area. Catastrophes such as these, are unpredictable. We understand that wetlands are fragile places that suffer greatly from floods; they should be monitored, both in the short and long term, in relation to changes in the environment so that conserving, maintaining, or rehabilitating actions can occur faster when needed.

**Author Contributions:** Conceptualization: M.S.N., A.A.P., D.C.B., and C.M.d.S.C.; Formal analysis: M.S.N. and A.A.P.; Funding acquisition: M.S.N., A.A.P., D.C.B., C.M.d.S.C., and S.C.S.; Investigation: M.S.N. and A.A.P.; Methodology: M.S.N. and A.A.P.; Resources: M.S.N., A.A.P., D.C.B., C.M.d.S.C., and S.C.S.; Visualization: M.S.N., A.A.P., D.C.B., C.M.d.S.C., and S.C.S.; Writing—original draft: M.S.N. and A.A.P. All authors have read and agreed to the published version of the manuscript.

**Funding:** This research was funded by Coordenação de Aperfeiçoamento de Pessoal de Nível Superior—Brasil (CAPES) grant number OO1, PDSE process number 8881.134251/2016-0, scholarship awarded to the first author. And was funded by NitroPortugal, H2020-TWINN-2015, a Coordination and support action n. 692331 and Fundação para a Ciência e a Tecnologia I.P. (FCT), Portugal (UIDB/00239/2020).

**Institutional Review Board Statement:** Not applicable.

**Informed Consent Statement:** Not applicable.

**Data Availability Statement:** No new data were created or analyzed in this study.

**Acknowledgments:** This study was financed in part by the Coordenação de Aperfeiçoamento de Pessoal de Nível Superior—Brasil (CAPES)—Finance Code 001. The authors would like to thank the Coordenação de Aperfeiçoamento de Pessoal de Nível Superior—CAPES, for the scholarship awarded to the first author, PDSE (Programa de Doutorado Sanduíche no Exterior) process number 88881.134251/2016-01. We thank the Gerpel/Unioeste (Grupo de Pesquisas em Recursos Pesqueiros e Limnologia) for providing us with the Chlorophyll-a and total P chemical analysis of water samples. We thank the divers, 6th Naval District of the Brazilian Navy, for the logistic support in the accomplishment of the collection of sediment, and Marcelo Bevilacqua Remor for supporting the sampling the water. The authors acknowledge NitroPortugal, H2020-TWINN-2015, a Coordination

and support action n. 692331 project for funding. We also thank the Forest Research Centre, a research unit funded by Fundação para a Ciência e a Tecnologia I.P. (FCT), Portugal (UIDB/00239/2020).

**Conflicts of Interest:** The authors have declared that no competing interests exist.

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
