# Peer review of "Variation of Diatoms at Different Scales in the Brazilian Pantanal Basin"

_water, doi:10.3390/w13060823_

Round 1

Reviewer 1 Report

General comments:

In this manuscript performed analyzed the structure of the diatom metacommunity in three permanent ponds. This manuscript provides some good information on diversity, aquatic habitat and connectivity not necessarily fish and benthic macroinvertebrate adaptability. This is an interesting issue for scientists and river managers. The manuscript is well-written and the idea is clear. However, the organizing the paragraph is awkward. Thus I recommend the manuscript “Major Revision.” In that sense, comparisons with existing results of relevant this manuscript may be more informatics and persuasive of the author’s findings. Some possible directions for improvement are listed below.

Specific comments:

(1) This also seems a very local study, with no implication elsewhere: to increase the scope of this manuscript, and not being once again a local case-study, it would be useful to point out here, why this study is important, i.e. how the findings of the present study can be applied elsewhere. This should be included in the last sentence of the abstract as well as in the Discussion and Conclusion. What is indeed the novelty of this study? Can this be applied elsewhere? You should focus on these messages to the readers, as it is what they want to really know.

(2) Please formulate clear objective, and further interest of this work. Is it intended to create a protocol that can be useful in many other projects of sediment transport, water quality, and habitat change of the aquatic species?

(3) It is a good idea to state what impacts you have on other fish species and whether you have any future plans.

(4) In the methods there are lacks of descriptions of the target species characteristics, sample sites, methods and techniques for sampling, how, when, where etc. this target species was sampled. Perhaps it would be good the authors providing some information about ecology of the target species (i.e. type of habitat, reproduction timings, tolerance to human pressures, etc.).

(5) Biologically, short- or long-term repetition of environmental parameter change must have the biggest impact on the aquatic species’ adaptability. Although the reviewer understands this situation as one stage of the research, it should be mentioned somewhere in the manuscript. In addition, do you have a monitoring plan to see if the response of the fish and benthic macroinvertebrate has any effect on the other water quality?

(6) Conclusions: in the present form this chapter is rather a summary than conclusions, describing the study processes instead of giving ‘take home messages’ from the results of the study for the reader.

(7) What is the research hypothesis?

Reviewer 2 Report

I have read with interest your paper on Variation of diatoms at different scales in the Brazilian Pantanal basin. While I suggest having a technical person reading and providing comments on this paper too, from my social sciences perspective I would suggest considering the following points to provide a more comprehensive analysis of the paper.

In Brazil, the issue of scale and of shred waters, as highlighted in the title and in the paper, is important. I would suggest therefore to include the issue of transboundary waters - although no need to explain too much in details about it - but we need to show awareness of this important aspect, which is mentioned as the Guarani Aquifer System for instance. How these are incorporated in the national directives (think of the different scales issue, and how waters are subject to different governance structures and dynamics in the case of shared groundwater - which are more locally used and decided upon - and surface waters; e.g. Guarani VS La Plata). I suggest reading and including the following two studies: 1)  da Silva, Luis Paulo Batista. "Production of scale in regional hydropolitics: an analysis of La Plata River Basin and the Guarani Aquifer System in South America." Geoforum 99 (2019): 42-53; 2) Hussein, H. (2018). The Guarani Aquifer System, highly present but not high profile: A hydropolitical analysis of transboundary groundwater governance. Environmental Science & Policy83, 54-62.

These elements would need to be liaised nicely with your discussion on scales, which now is not as elaborated upon as you could instead. Look in particular at the discussion on scales in water in Brazil developed by the Brazilian researcher Luis Paulo de Silva (see above).

For more work on trasboundary and scales in Latin America, you may also want to look at the work of Francesco Sindico. 

I hope this helps. 

Round 2

Reviewer 1 Report

Congratulations, now all issues were reflected in the revised manuscript.

Reviewer 2 Report

Much improved